# Hepatitis C Virus Screening among Baby Boomers: The Positive Benefits of Health Education and Outreach during the COVID-19 Pandemic

**DOI:** 10.3390/healthcare11030302

**Published:** 2023-01-19

**Authors:** Carlos Lopez Bray, Richard Taylor, Naomi Tamez, Wesley Durkalski, John R. Litaker

**Affiliations:** 1Office of Population Health, Sendero Health Plans, Inc., Austin, TX 78741, USA; 2Undergraduate Public Health Program, The University of Texas at Austin, Austin, TX 78712, USA; 3Former Chief Executive Officer, Sendero Health Plans, Inc., Austin, TX 78741, USA; 4Office of Population Health and Science, The Litaker Group, LLC, Austin, TX 78716, USA

**Keywords:** Hepatitis C Virus screening, community-based, COVID-19, prevention

## Abstract

In 2019, 2020, and 2021, Sendero Health Plans, an ACA health insurance company, implemented Hepatitis C Virus (HCV) health education and outreach screening campaigns. Chi-square goodness-of-fit and tests of independence were performed to assess and compare the uptake of HCV screening among baby boomers in 2019, 2020, and 2021. In 2019, 2020, and 2021, 17.9% (308/1,718), 10.9% (93/852), and 8.5% (37/435) of eligible members were screened, respectively. Individuals were more likely to be screened for HCV in 2019 than in 2020 and 2021 (*p* < 0.0001). In 2019, 2020, and 2021, 39.9%, 26.9%, and 48.6% of annual screenings occurred during the health campaign months, respectively. Annual HCV screening rates were lower during the COVID-19 pandemic period than in the pre-pandemic cohort. However, screening rates during the months of outreach and education contributed to nearly 50% of annual screenings in the pandemic year 2021, thus representing a positive impact on preventive screening uptake despite the pandemic. Missed screening opportunities affect HCV transmission, diagnosis, and treatment. Yet, health education and outreach continue to work, even during a pandemic.

## 1. Introduction

Globally, 58 million people are estimated to be chronically infected with the Hepatitis C Virus (HCV), with an additional 1.5 million acquiring the infection every year [1]. In the United States, HCV is the most common chronic bloodborne pathogen, affecting an estimated 2.4 million people [2]. According to the United States Centers for Disease Control and Prevention (CDC)*,* people born between 1945 and 1965 (baby boomers) have historically been the most impacted cohort in the United States, with an HCV prevalence five times higher than other adults [3]. To address this disease on the global stage, the World Health Organization (WHO) released the first global health sector strategy on viral hepatitis, with the goal of eliminating hepatitis, including HCV, by 2030 through community interventions, infection prevention, disease detection, and effective treatment [4]. In 2013, the United States Preventive Services Task Force (USPSTF) published guidelines recommending a one-time HCV screening test for members of the baby boomer population; these guidelines were later revised and expanded in 2020 to include all adults 18 to 79 years of age [5]. Although HCV screening among baby boomers has increased within the last decade, it is estimated that 50% of all HCV carriers are unidentified [6,7]. Health insurance companies in the United States providing coverage under the Affordable Care Act (ACA) have a statutory obligation to provide coverage for USPSTF items with a recommendation rating of “A” or “B”. HCV screening is a USPSTF “A” recommendation.

In 2020, the COVID-19 pandemic reached the United States. In the early stages of the pandemic, the CDC and the Centers for Medicare and Medicaid Services (CMS) recommended postponing all non-essential medical services, including preventive services, to conserve healthcare resources and to prevent exposure to the severe acute respiratory syndrome coronavirus 2 (SARS-CoV-2) [8]. In addition, the governor of Texas issued executive orders in March 2020 to postpone all “surgeries and procedures that [were] not immediately medically necessary” [9]. These restrictions facilitated a shift in medical services from an office-based setting to a virtual-based setting, limiting access to in-person medical and preventive services like HCV screening [10]. Reduced access to non-urgent in-person care leads to decreased uptake of preventive screening services. A reduction in HCV screening may result in missed opportunities for diagnosis and treatment, leading to the progression of HCV-related liver fibrosis or cirrhosis. In addition, an infected individual with an unknown status remains capable of infecting others. 

In 2017 and 2018, Sendero Health Plans, Inc. (Sendero), a non-profit community-based health maintenance organization in Austin, Texas, providing health insurance coverage under the Patient Protection and Affordable Care Act (ACA), created a series of education and outreach campaigns to increase preventive screenings for breast cancer, diabetic retinopathy, and hepatitis C [11,12,13]. These campaigns occurred prior to the advent of the COVID-19 pandemic. However, from a policy perspective, Sendero continued to work with its members to prioritize preventive services during the pandemic years 2020 and 2021. The purpose of this study is to assess and compare the uptake of HCV screening among the baby boomer cohort in the pre-pandemic year 2019 to the pandemic years 2020 and 2021 by a non-profit, community-based health plan.

## 2. Materials and Methods

### 2.1. Health Education and Outreach Campaigns

In 2019, 2020, and 2021, Sendero designed and implemented annual health education and outreach campaigns to encourage eligible adult members in Central Texas to obtain an HCV screening. The annual HCV campaigns were initiated in October 2019, November 2020, and September 2021. Outreach for each annual cohort was based on prevailing USPSTF recommendations at the time; the 2019 cohort targeted the baby boomer population (born between 1 January 1945 and 31 December 1965), while the 2020 and 2021 cohorts targeted adults aged 18 to 79 years, with the latter two cohorts including the expanded USPSTF age criteria. 

Members were eligible to receive an HCV screening test if they met the age inclusion criteria set by the USPSTF, and if there was no evidence of a previous HCV screening in the Sendero claims database as evidenced by documentation of one of the following Current Procedural Terminology (CPT) codes: 86803 (HCV antibody screening); 86804 (Hepatitis C antibody); 87521 (definitive diagnosis of hepatitis C by amplified probe technique); 87522 (HCV quantification); 87902 (HCV genotype) or the Healthcare Common Procedure Coding System (HCPCS) code G0472 at any time prior to the annual campaign start date. 

In each of the three cohort years, the Sendero health education and outreach campaign included a personalized letter encouraging the member to take advantage of free HCV screening before the end of the calendar year. The letter noted that screening was a covered benefit with no additional out-of-pocket cost to the member, provided educational information about HCV and the benefits of early detection, offered assistance to members needing help scheduling an appointment, and offered a USD 25.00 gift card to a local grocery merchant if the screening test was completed on or before 31 December of the cohort calendar year. The timeframe of the screening campaigns varied by year due to operational considerations. The 2019 campaign occurred from 18 October through 31 December 2019, the 2020 campaign occurred from 9 November through 31 December 2020, and the 2021 campaign occurred from 10 September through 31 December 2021.

### 2.2. Data Analysis

Inclusion criteria for data analysis included individuals born between 1945 and 1965 (i.e., the baby boomers) because this cohort was consistently included in all three cohorts, and individuals who had obtained an HCV screening test based on CPT codes 86803, 86804, 87521, 87522, and 87902 and HCPCS code G0472. Additionally, in order not to violate the statistical assumption of independence of observations, inclusion was limited to individuals who were eligible to be screened during a single year. 

Chi-square (χ2) goodness-of-fit tests with appropriate degrees of freedom χ2[degrees of freedom] were performed to assess the annual distribution of screenings by month in calendar years 2019, 2020, and 2021. The threshold to determine statistical significance was set at α = 0.05. *p*-values are reported to compare the uptake of HCV screening tests among members of the baby boomer birth cohort for each calendar year. 

An overall chi-square test of independence was performed to understand if the year screened was independent of the proportion of people who chose to be screened. The χ2 with appropriate degrees of freedom and *p*-value are reported. Additionally, prevalence ratios (PR), *p*-values, and 95% confidence intervals (CI) were used to understand changes in screening rates across time and during the first two years of the pandemic. The threshold to determine statistical significance for the chi-square χ2 tests of independence was set at α = 0.05. 

## 3. Results

### 3.1. Annual Screening Rate

In 2019, 1718 unique members were eligible to receive a one-time HCV screening test; 308 (17.9%) were screened in the pre-pandemic year 2019. Of these 308 members, 163 (52.9%) were female. The chi-square goodness-of-fit test is statistically significant (χ^2^_[11]_ = 111, *p* < 0.0001).

In 2020, 852 unique members were eligible to receive a one-time HCV screening test; 93 (10.9%) were screened in the pandemic year 2020. Of these 93 members, 43 (46.2%) were female. The chi-square goodness-of-fit test is statistically significant (χ^2^_[11]_ = 24.7, *p* < 0.05).

In 2021, 435 unique members were eligible to receive a one-time HCV screening test; 37 (8.5%) were screened that year. Of these 37 members, 17 (46%) were female. The chi-square goodness-of-fit test is statistically significant (χ^2^_[11]_ = 31.9, *p* < 0.001).

### 3.2. Annual Comparisons

Evidence suggests there is a relationship between the number of eligible individuals screened and the calendar year (χ^2^_[2]_ = 37.54, *p* < 0.0001). In the pre-pandemic year 2019, individuals were 1.64 times more likely to be screened for HCV than in the pandemic year 2020 (PR = 1.64; *p* < 0.0001; 95% CI: 1.32, 2.04). Similarly, in the pre-pandemic year 2019, individuals were 2.11 times more likely to be screened than in the pandemic year 2021 (PR = 2.11; *p* < 0.0001; 95% CI: 1.52, 2.91). There is no statistical difference in HCV screening between the pandemic year 2020 and the pandemic year 2021 (PR = 1.28; *p* = 0.17; 95% CI: 0.89, 1.85).

### 3.3. Uptake during the Health Education and Outreach Campaigns 

In the pre-pandemic year 2019, 39.9% (123 of 309) of annual HCV screenings occurred during the health education and outreach campaign period from 18 October through 31 December 2019 (Figure 1). In the pandemic year 2020, 26.9% (25 of 93) of annual HCV screenings occurred during the campaign period from 9 November through 31 December 2020 (Figure 2). In the pandemic year 2021, 48.6% (18 of 37) of annual HCV screenings occurred during the campaign period from 10 September through 31 December 2021 (Figure 3). Table 1 presents the monthly count of HCV screenings for the study cohort in each calendar year.

## 4. Discussion

This study demonstrates two important concepts associated with the delivery of preventive services, like HCV screening, during a pandemic.

Strict stay-at-home orders and concerns about infection, morbidity, and mortality with COVID-19 reduced the uptake of HCV-related preventive screening services during a pandemic;Health education and outreach continued to have a positive impact on preventive screening services despite the ongoing pandemic.

The infectious nature of the SARS-CoV-2 virus required strong mitigation measures until a vaccine could be developed, manufactured, and distributed. Strict stay-at-home orders and medical service restrictions were implemented at the national, state, and local levels during the COVID-19 pandemic and disrupted the delivery of in-person preventive medical services to conserve sufficient resources for the COVID-19 response. This reduced the uptake of many non-emergent and preventive health services, including HCV screening. Our findings demonstrate a statistically significant decrease in HCV screening during the pandemic’s first and second years, as compared to the immediately preceding pre-pandemic year. Similar findings were reported in a study analyzing the volume of laboratory tests for the period of March–July 2020 as compared to a similar period in 2018 and 2019. The authors report that “the decrease in HCV antibody testing during the pandemic year was statistically significant (*p* < 0.001)” [14]. Additional studies analyzing the uptake of preventive exams for screening mammography and routine medical care report a similar impact of decreased utilization during the pandemic [15,16]. 

Our study adds to this body of knowledge in several ways. For example, as a health insurance provider, Sendero has the benefit of identifying both the eligible population (denominator) and the individuals who obtained the screening test (numerator). Unlike other studies that rely on numerator data only and focus solely on volume [14], our data considers both the eligible population for the screening exam and the population that obtains the exam. We show that uptake decreased from 17.9% in the pre-pandemic year of 2019 to 10.89% and 8.5% in the pandemic years of 2020 and 2021, respectively. We are also confident that we did not violate the independence of observations with our data, thus presenting an accurate picture of uptake across the cohort years. 

A second benefit of our study is that it assesses the impact of education and outreach on the pre-pandemic and pandemic cohorts. This is important because education and outreach are core components of public health, and studies have demonstrated their effectiveness in improving service utilization across a variety of clinical diseases [17,18,19]. Ours is the first study to report on the impact of education and outreach activities for HCV screening during a pandemic and to compare it to a pre-pandemic cohort of like individuals. Importantly, Sendero Health Plans continued to provide health education and outreach for diseases that had an evidence-based screening protocol. In the pre-pandemic year 2019, the proportion of individuals screened during the months when health education and outreach occurred was 39.6% of the annual total. In the pandemic year 2020, the proportion decreased to 26.9%, but an increase in HCV screening occurred in the education and outreach months of November and December, despite the fourth pandemic wave, the continued use of telemedicine, the lack of vaccines, and the fact that baby boomers were at increased risk of COVID-19 infection due to their age (55–75 years). In the pandemic year 2021, the proportion rebounded to 48.6%, a value greater than even before the pandemic. This is an important yet unexpected finding that shows the benefit of health education and outreach, even during a secular event that disrupted the health and medical system for nearly two years. Data from 2021 may also indicate an increase in service uptake due to pent-up demand for previously missed medical services.

### 4.1. Public Health Implications 

Our study demonstrates that HCV screening rates were lower during the COVID-19 pandemic period than in the pre-pandemic cohort. Missed screening opportunities can delay the diagnosis of HCV and allow the transmission to continue in the population by preventing individuals from undergoing effective direct-acting antiviral treatment [20]. Additionally, missed treatment opportunities contribute to the direct and indirect costs associated with HCV infection, which in the United States are estimated to be greater than USD 10 billion annually [21]. While it is still too early to measure the health outcomes of missed HCV screening during the COVID-19 pandemic, we would expect an impact on the HCV clinical course as has been seen in other diseases. For example, a post-pandemic study on breast cancer demonstrated that a 2-month delay in screening mammography in 2020 resulted in cancers being diagnosed at more advanced stages once screenings resumed [22]. Another study predicted an increase in advance-staged diagnoses and cancer-related deaths in 2020–2029 for breast cancer due to a 3-month delay in screenings during the COVID-19 pandemic [23]. While we do not yet have data to show the impact of delayed screenings on HCV-related fibrosis and hepatitis, we would expect an increase in HCV morbidity and mortality over time consistent with what has been seen for other diseases for which preventive screening was available but unused during the pandemic. 

Although COVID-19 transmission control became a public health priority worldwide, the development of pandemic-response guidelines should consider the potential long-term consequences of diseases that have been neglected during the pandemic event and acknowledge the risk of implementing public health response strategies that enable a one-disease health system, even if only temporarily. Indeed, while COVID-19 was the focus of public health, it was not the sole disease occurring during this time period.

### 4.2. Limitations

This study has a few known limitations. First, the eligible baby boomer population enrolled in commercial and ACA health insurance plans decreased each year as they became eligible for Medicare enrollment. Second, our data is limited to when a person is a Sendero member, and we did not conduct a medical record review for periods of time prior to enrollment with us. Third, the health education and outreach campaigns took place during the later months of the year. Several nationally recognized holidays occur during November and December, which may have limited opportunities for members to schedule appointments. Lastly, because the study was conducted on an insured population, a bias may manifest in potentially higher uptake, as individuals with health insurance may have a cognitive bias toward obtaining a health screening service. 

## 5. Conclusions

Disease prevention is a cornerstone of population health. In 2019, 2020, and 2021 Sendero Health Plans provided education and outreach to encourage members to be screened for HCV, in support of nationally established evidence-based preventive service recommendations and global strategies. While education and outreach are part of the ongoing commitment to improving the health of Sendero members, they became critically important during the pandemic years of 2020 and 2021, during which, strict stay-at-home orders, conservation of medical resources, and general fear and anxiety prevented individuals from seeking care for non-essential services, including preventive screenings. The data presented in this manuscript supports the notion that utilization patterns changed, with a decrease in HCV screening reported in pandemic years, yet with a continued robust impact of health education and outreach during the health campaign months. 

We will see future public health emergencies and pandemics. The lessons learned from this pandemic, both on the COVID-19 front and non-COVID-19 front, should be embraced. One lesson is that pandemic-focused response and mitigation policies and strategies, enabling a one-disease health system, should consider the risks and long-term consequences of shifting focus away from preventive care. Secondly, health insurance providers play a key role by supporting the uptake of nationally recommended preventive services during a pandemic and can align community-based efforts with national and global strategies. 

## Figures and Tables

**Figure 1 healthcare-11-00302-f001:**
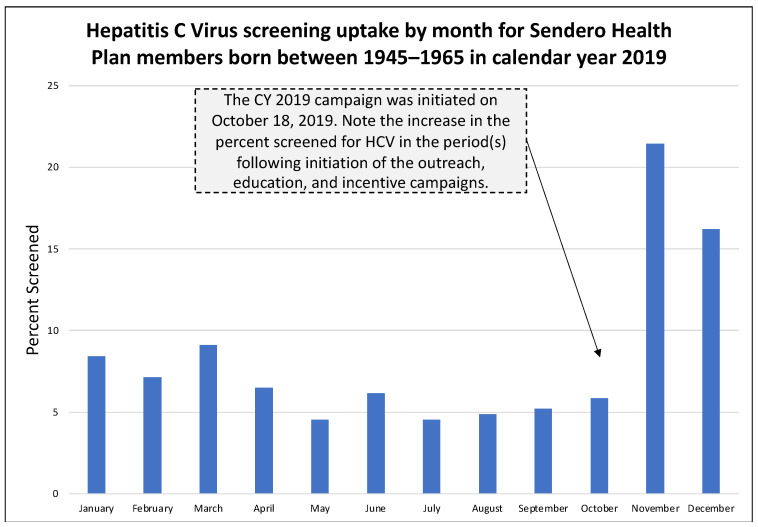
Hepatitis C Virus uptake by month for Sendero members born between 1945–1965 in calendar year 2019.

**Figure 2 healthcare-11-00302-f002:**
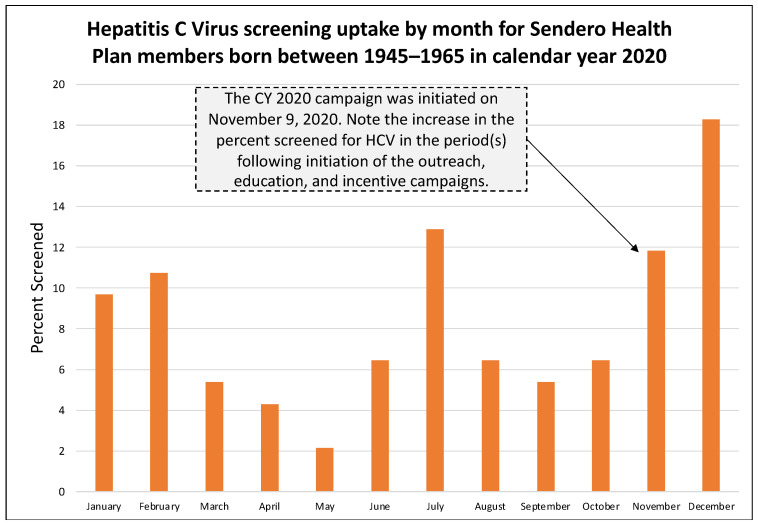
Hepatitis C Virus uptake by month for Sendero members born between 1945–1965 in calendar year 2020.

**Figure 3 healthcare-11-00302-f003:**
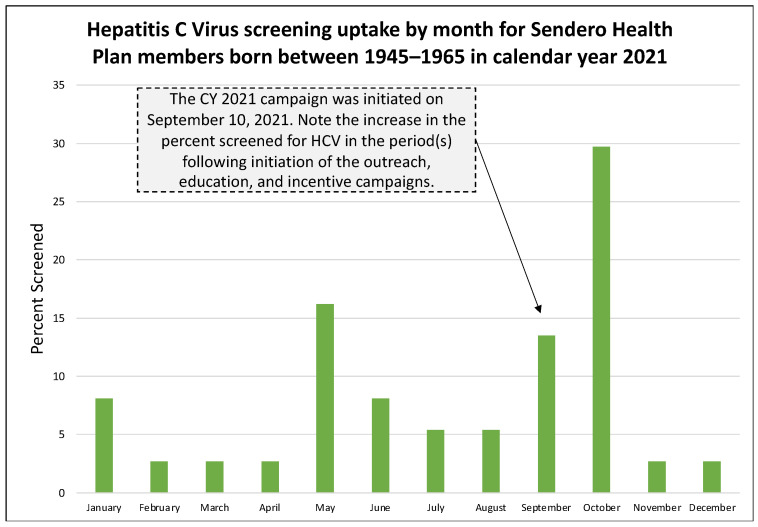
Hepatitis C Virus uptake by month for Sendero members born between 1945–1965 in calendar year 2021.

**Table 1 healthcare-11-00302-t001:** Count of Hepatitis C Virus (HCV) Screenings for the Study Cohort in Calendar Years 2019–2021, by Month.

*Year*	*January*	*Februay*	*March*	*April*	*May*	*June*	*July*	*August*	*September*	*October*	*November*	*December*	*Total Screened*	*Total Eligible*
2019	26	22	28	20	14	19	14	15	16	18	66	50	308	1718
2020	9	10	5	4	2	6	12	6	5	6	11	17	93	852
2021	3	1	1	1	6	3	2	2	5	11	1	1	37	435

## Data Availability

The data that was generated and analyzed in this study are available from the corresponding author upon request.

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
