# Peer review of "Hepatitis C Virus Screening among Baby Boomers: The Positive Benefits of Health Education and Outreach during the COVID-19 Pandemic"

_healthcare, 2023, doi:10.3390/healthcare11030302_

Round 1

Reviewer 1 Report

Carlos Lopez Bray et al. discuss in their article the uptake of hepatitis C virus educational campaign and preventive screening tests issued from a health insurance company in Texas / USA before and during the nation-wide COVID 19 pandemic measures. 

The result was that in pre-pandemic times it was nearly double so likely that individuals let themself screen for hepatitis C infection than during the pandemic. This is in agreement with the use of other other preventive measures in the health sector - like screens for a mama carcinom.  

At the same time, an educational campaign was more significant for convincing people to do the screening test. Of those who did take the test a higher percentage did this in context of the educational campaign. 

The article expresses its purpose well in its title. Its abstract summarises the findings appropriately. The data in the article support the statements of the authors and the figures highlight the findings. And the conclusions the authors formulate are appropriate. 

The main concern I have is that the findings are not significant enough to justify the publication of an article. But this is up to the editor to decide. 

Reviewer 2 Report

The study entitled “Hepatitis C virus prevention: the positive benefits of health education and outreach during the COVID-19 pandemic” is very important. Author reach to the baby boomers for the screening of HCV. Which is very important step to reach Hepatitis C elimination.

I have few suggestions for the improvement of paper.

Title: there is a need to improve the title of the paper. The first portion of title “HCV prevention” is not suitable. it can be replaced with

“Hepatitis C virus screening in baby boomers: the positive benefits of health education and outreach during the COVID-19 pandemic”

or

“Finding the missing millions: the positive benefits of health education and outreach during the COVID-19 pandemic”.

If it is possible for you, please add details, how many of the study participants were HCV positive and how many of them were linked to HCV treatment program. This will add great value to your study.

HCV is a global health problem. Please start your paper with global burden of HCV. The best paper for global burden of HCV is from Polaris, USA, published in Lancet Gastroenterology & Hepatology in 2022.

https://www.sciencedirect.com/science/article/abs/pii/S2468125321004726

In introduction, please add the details of WHO Global health Sector strategy on Viral hepatitis and how much progress is achieved. You may cite following paper.

Progress on global hepatitis elimination targets, published in World Journal of Gastroenterology.

https://pubmed.ncbi.nlm.nih.gov/35068864/

Globally, majority of HCV cases are undiagnosed and you did this work in USA. World Hepatitis Alliance has started a campaign to named “Find the Missing Millions” to screen the millions of HCV cases who are undiagnosed. Please write about this campaign in your research paper, because your work is directly linked with this campaign and I also suggested you the title similar to that.

You may get some missing millions information from this research paper.

https://www.ncbi.nlm.nih.gov/pmc/articles/PMC6262254/

This study is very important and it should published with the addition of above suggestions.

Reviewer 3 Report

In this study, the authors tried to investigate the impact of the COVID-19 pandemic on the uptake of HCV screening among baby boomers as well as the annual screening rate. Although the topic is interesting, I have several concerns that should be addressed, as follows:

1-      The major concern is the limitation of the screening to Sendero members, which may introduce selection bias into the results. Also, the sample size appears to be relatively small.

2-      There is some heterogeneity in the studied cohorts among different years since the 2019 cohort targeted the baby boomer population (born between January 1, 1945, and December 31, 1965), while the 2020 and 2021 cohorts targeted adults aged 18 to 79 years. So the results should be adjusted for age.

3-      The conclusion is unnecessarily long and needs to be focused on the main study findings.

Reviewer 4 Report

Dear authors, 

your study is quite interesting and provides some insights about screening procedures, highlighting the importance of stressing education processes. 

However, I feel that its novelty is overall poor. 

I feel that the word prevention in the title does not reflect what you actually did (screening). I would possibly change it. I would discuss more the importance of detecting people with chronic hepatitis C for the possibility to treat them tailoring therapy and preventing the traditionally known complications, and the economic consequences. On this purpose, please see:

- Marcellusi et al, 2019,

- Marascio et al, 2019 

Best regards

Round 2

Reviewer 4 Report

None